# Recent Advances in Decellularized Extracellular Matrix-Based Bioinks for 3D Bioprinting in Tissue Engineering

**DOI:** 10.3390/ma16083197

**Published:** 2023-04-18

**Authors:** Man Zhe, Xinyu Wu, Peiyun Yu, Jiawei Xu, Ming Liu, Guang Yang, Zhou Xiang, Fei Xing, Ulrike Ritz

**Affiliations:** 1Animal Experiment Center, West China Hospital, Sichuan University, Chengdu 610041, China; 2West China Hospital of Stomatology, Sichuan University, Chengdu 610041, China; 3LIMES Institute, Department of Molecular Brain Physiology and Behavior, University of Bonn, Carl-Troll-Str. 31, 53115 Bonn, Germany; 4Orthopedic Research Institute, Department of Orthopedics, West China Hospital, Sichuan University, Chengdu 610041, China; 5Department of Orthopaedics and Traumatology, Biomatics Group, University Medical Center of the Johannes Gutenberg University, Langenbeckstr. 1, 55131 Mainz, Germany

**Keywords:** bioprinting, bioinks, tissue engineering, decellularization, tissue repair, decellularized extracellular matrix

## Abstract

In recent years, three-dimensional (3D) bioprinting has been widely utilized as a novel manufacturing technique by more and more researchers to construct various tissue substitutes with complex architectures and geometries. Different biomaterials, including natural and synthetic materials, have been manufactured into bioinks for tissue regeneration using 3D bioprinting. Among the natural biomaterials derived from various natural tissues or organs, the decellularized extracellular matrix (dECM) has a complex internal structure and a variety of bioactive factors that provide mechanistic, biophysical, and biochemical signals for tissue regeneration and remodeling. In recent years, more and more researchers have been developing the dECM as a novel bioink for the construction of tissue substitutes. Compared with other bioinks, the various ECM components in dECM-based bioink can regulate cellular functions, modulate the tissue regeneration process, and adjust tissue remodeling. Therefore, we conducted this review to discuss the current status of and perspectives on dECM-based bioinks for bioprinting in tissue engineering. In addition, the various bioprinting techniques and decellularization methods were also discussed in this study.

## 1. Introduction

In recent years, three-dimensional (3D) bioprinting has been widely utilized as a novel manufacturing technique by more and more tissue engineering researchers to construct various organ or tissue substitutes with complex architectures and geometries [1,2,3]. Moreover, 3D bioprinting, integrating regenerative medicine, 3D printing, biomaterials, and tissue engineering can construct tissue or organ substitutes by accurately bioprinting various bioinks layer by layer via a computer-controlled dispensing system [4,5,6]. In the field of tissue engineering and regenerative medicine, 3D bioprinting technology utilizes hydrogels as bioinks to load various cells to fabricate tissue substitutes [7]. Compared with other biomaterial manufacturing techniques, 3D bioprinting can achieve precise control and the distribution of different bioactive factors, such as stem cells, small molecules, growth factors, miRNA, and drugs, which play a vital role in modulating the process of tissue formation and remodeling [8,9,10,11]. More and more tissue engineering researchers have successfully utilized various tissue or organ substitutes constructed by 3D bioprinting technology to replace or repair damaged tissues and organs, such as bone, cartilage, and skin [12,13,14,15].

During the fabrication of tissue substitutes by 3D bioprinting, the bioink, as an essential part, is utilized to encapsulate and support various cells, such as stem cells [16,17]. An extracellular matrix (ECM) is a network structure formed by various proteins and polysaccharides distributed in the extracellular space [18]. The unique network structure of the ECM imparts specific mechanical properties to tissues and organs and provides the necessary cellular microenvironment for cell survival. Furthermore, the collagens, growth factors, and glycosaminoglycans (GAGs) of the ECM can modulate cell proliferation, signaling, attachment, and tissue development [19,20,21]. As a popular biomaterial derived from native tissues in regenerative medicine, the decellularized extracellular matrix (dECM) not only preserves the main ECM components but also possesses low immunogenicity after removing lipids, nucleic acids, membranes, cytoplasmic structures, and other immunogenic substances [22,23,24]. In 3D bioprinting, more and more researchers are developing the dECM as a novel bioink for the construction of tissue substitutes [25,26,27]. Compared with other biomaterials, the dECM possesses inner structural integrity and tensile strength similar to native tissues [28,29]. In addition, the various ECM components in dECM-based bioink can regulate cellular functions, modulate the tissue regeneration process, and adjust tissue remodeling [30,31]. In addition, the dECM has been utilized to fabricate tissue constructs for diseased tissue models, drug screening, tissue regeneration, pathology studies, and targeted drug and cell delivery [32]. Therefore, we conducted this review to discuss the current status of and perspectives on dECM-based bioinks for bioprinting in tissue engineering. 

## 2. Bioprinting Technology of Decellularized Extracellular Matrix

According to the working principle, bioprinting based on dECM bioinks can be classified into inkjet-based bioprinting, laser-assisted bioprinting, stereolithography-based bioprinting, and extrusion-based printing. Figure 1 shows the four bioprinting methods [33,34,35]. 

### 2.1. Inkjet-Based Bioprinting

Inkjet-based bioprinting deposits micro- or nano-liter bioink drops containing cells onto the platform layer by layer to stack target 3D objects [36,37,38]. During the process of bioprinting, a piezoelectric actuator or thermal change deforms the printing heads to inkjet the bioink and produces droplets. In the inkjet-based bioprinter, piezoelectric heads contain elements composed of substances sensitive to charge, such as crystals and ceramics. When a pulse voltage is supplied, the element bends back, forcing a precise amount of bioink from the ink cartridge to the substrate [39]. For inkjet-based bioprinting, the viscosity and surface tension determine the droplet size and deposition rate. Furthermore, the printing pathway and bioink deposition process during the process of bioprinting can be modulated by precisely changing the driving voltage and vibration frequency [25]. The vibration frequency of the electric signal activating the actuator determines the speed of droplet deposition [40,41]. In addition, inkjet-based bioprinting can enable the printing of single cells by reducing the volume of bioink [42]. Furthermore, the printing speed of inkjet-based bioprinting can be adjusted arbitrarily, which greatly promotes the further application of inkjet-based bioprinting [43]. Despite the many advantages of inkjet-based bioprinting, it is still limited by problems such as nozzle clogging, especially when printing high-cell-density or high-viscosity bioinks [25].

### 2.2. Laser-Assisted Bioprinting

During the process of laser-assisted bioprinting (LAB), laser energy is used to volatilize a sacrificial layer and propel a payload to a receiving substrate [44]. Differing from other bioprinting techniques, LAB is a form of nozzle-free bioprinting. A pulsed laser beam, a laser-absorbing layer, and a receiving substrate comprise the LAB system. In addition, the receiving substrate acts as a platform for droplet deposition and cell adhesion [45]. The LAB printing principle mainly generates high-pressure bubbles by laser irradiation of the ribbon layer, which, in turn, can push the bioink to produce droplets, which are finally deposited on the receiving plate [46]. The printing effect of LAB is affected by many factors, such as the viscosity of the bioink, the surface tension of the accepted substrate, and the laser intensity [47]. Compared to inkjet-based bioprinting, LAB enables the printing of high-viscosity or high-cell-density bioinks. However, the main disadvantages of laser-assisted bioprinting include the cell damage induced by laser energy and the high costs of constructing a laser-assisted bioprinting system [48,49].

### 2.3. Stereolithography-Based Bioprinting

As the earliest commercialized 3D bioprinting technology, stereolithography-based bioprinting was invented by Charles in 1985 [50]. In recent years, stereolithography-based bioprinting has been widely used in bio-fabrication for drug screening, pathophysiological research, and disease modeling due to its high-precision characteristics [51]. In addition, during the fabrication procedure, stereolithography-based bioprinting does not apply shear forces to cells, which would protect the cell viability [52]. In the field of biomedicine, stereolithography-based bioprinting was first utilized to fabricate a highly accurate skull model for reconstructive surgery [33]. Stereolithography-based bioprinting follows the features of ordinary 3D bioprinting layer-by-layer printing. When concentrated ultraviolet light is shone on each layer of bioink, the bioink at the place irradiated by the light receives the energy provided by the light and forms covalent bonds with adjacent molecular chains [53], showing the curing effect on the macroscopic level. When a spot is cured, the beam is then moved until the entire layer area is cured [54]. The disadvantage of stereolithography-based bioprinting is that the bioink must be transparent, otherwise light will not pass through the bioink evenly and the material will cure unevenly. 

### 2.4. Extrusion-Based Bioprinting

Extrusion-based bioprinting can be classified into mechanical and pneumatic systems. In addition, mechanical systems can be divided into piston and screw types. The printing principle of extrusion-based bioprinting mainly involves continuously extruding the bioink, which then flows out through the nozzle and forms filaments that are stacked into the target shape [25,40]. In addition, extrusion-based bioprinting can be used for high-viscosity or high-cell-density bioink printing compared to inkjet-based bioprinting [55]. Furthermore, extrusion-based bioprinting provides a varied selection of biomaterials [56]. The pneumatic system employs a clean compressed air source to generate stable air pressure to the bioink in the cartridge. As for the mechanical system, the piston or screw module exerts forces on the bioink. The piston-driven system enables the arbitrary regulation of the bioink flow. In addition, screw-based systems are more suitable for high-viscosity or high-cell-density bioink printing compared to piston-driven systems [57]. In the printing process of extrusion-based bioprinting, cells loaded in the bioink are subjected to shear stress as they pass through the nozzle, which greatly affects the cell activity in the later application. Researchers can reduce the effect of shear stress on cell activity by changing the viscosity of the bioink and the size of the nozzle. In addition, in the process of extrusion-based bioprinting, researchers can also regulate the cell activity encapsulated in a bioink by varying the printing temperature and printing time [25]. Compared to other bioprinting technologies, extrusion-based bioprinting can increase the printing speed of a bioink by increasing the diameter of the nozzle. In addition, extrusion-based bioprinting is better suited for the construction of large volumes of tissue or organ substitutes [40]. However, apart from these advantages, the relatively low resolution caused by shear stress and the poor cell viability caused by the detrimental effects of the shear damage owing to pressure or mechanical force need to be resolved [58,59,60,61].

## 3. Bioactive Molecules in the Decellularized Extracellular Matrix

The dECM is mainly composed of the main ECM, which includes collagen, elastin, fibronectin, matricellular proteins, laminin, and other extracellular macromolecules [62,63,64]. Although the components of the ECM vary in different tissues, the main components are generally classified into two types, fibrous proteins and glycoproteins [25,48,65,66,67]. The fibrous proteins mainly include elastin and various collagens, while the glycoproteins consist of laminin, fibronectin, and proteoglycans. These secreted macromolecules possess different functions (e.g., promoting cell adhesion, participating in cell signaling, regulating protein complexes). Fibrous proteins, including collagen and elastin, are structural proteins and are distributed in most soft tissues [25]. In addition, fibrous proteins provide tensile strength, influence cell types’ disposition, and connect the framework of the tissue/organ [68] (Figure 2). In addition, as a protein with high resilience, elastin supplies elasticity, alters the mechanical properties, and increases hemocompatibility [69,70,71]. Glycoproteins provide tissue compressibility and turgidity and promote the transport of growth factors [69,70,71]. Take heparan sulfate proteoglycans (HSPGs) as an example. HSPGs with highly negative charges can bind to various receptors and regulate cellular processes, including cell growth and migration [72,73,74]. Adhesion glycoproteins, mainly including laminin and fibronectin, can bind to various structural molecules in the ECM to regulate the network strength of the ECM. In addition, adhesion glycoproteins regulate intercellular, cellular, and extracellular matrix signaling. Moreover, various biological activities of cells, including proliferation and migration, are also regulated by adhesion glycoproteins [75]. The functions of integrins include adjusting cellular behavior, participating in tissue repair, and remodeling [76]. 

Directing the stem cells encapsulated in bioinks or scaffolds to differentiate into target cells is still challenging because the bioink’s loading cells lack specific cell inductors [77]. The ECM can act as a dynamic microenvironment, or a natural niche, to modulate the fate and cellular behavior of stem cells encapsulated in dECM-based bioinks [78]. Figure 3 shows the role of the ECM acting as the cellular niche of stem cells [77]. In general, the function of the ECM can be classified into three types: biomechanical signaling, biochemical signals, and dynamic remodeling. Biomechanical signaling indicates the 3D structure composed of the ECM and the physical properties of the ECM [77]. The mechanisms by which biomechanical signaling regulates multicellular interactions, morphogenesis, and cellular behaviors include the spatial organization of the ECM, direct cellular binding, the separation between distinct structures, and the regulation of growth factors and cytokines [25,79]. In terms of biochemical signals, the bioactive agents of the ECM derived from tissues and organs are different. Various protein adhesion domains in the ECM can bind various bioactive factors. In addition, discrete cellular responses can be triggered by the extension of chemical cues among ECM-surrounding cells [25,79]. As for dynamic remodeling, in order to respond to environmental stimuli, the ECM is always in a dynamic balance to maintain homeostasis. 

## 4. Construction Methods of Decellularized Extracellular Matrix

To eliminate the immune reaction of cellular components, decellularization acts as a vital step in the fabrication of the dECM. In contrast, the core of decellularization is removing cells and major histocompatibility complexes from tissues and organs while retaining natural ECM structures and components [80]. Successful decellularization can effectively minimize the occurrence of immune rejection and preserve the biochemical composition and inner structural integrity, which provides seed cells with necessary clues for tissue regeneration [81]. However, during the decellularization process, various decellularization agents or techniques will influence the natural, native composition, and inner structure of the ECM inevitably. Therefore, to optimize the decellularization process, different tissues need to apply different schemes to ensure effective decellularization according to the difference in cell and matrix density [82]. To date, different methods have been used to remove cellular and nuclear components from the organ or tissue. The decellularization techniques can be classified into chemical, physical, and biological methods [83,84,85]. Figure 4 shows the common decellularization methods. 

### 4.1. Chemical Methods

Chemical methods mainly apply chemical agents for decellularization, such as acids and bases, hyper- or hypo-tonic solutions, ionic or non-ionic detergents, and alcohols, to disrupt cellular membranes and improve the hydrolytic degradation of biomolecules, so as to achieve decellularization [86,87]. Acid−base methods utilize various acidic or base agents to remove various nuclear residues and cellular components of the ECM [88]. Meanwhile, acidic agents for decellularization mainly include peracetic acid (PAA) and sulfuric acid. PAA, as a standard disinfecting and decellularization agent, can remove the nuclear residues and retain the essential growth factors [89]. Hodde et al. used a mixture of peracetic acid and aqueous ethanol to decellularize a porcine small intestine immersed in a virus solution [90]. The results demonstrated that the mixture of peracetic acid and aqueous ethanol could effectively kill the virus and achieve decellularization for further xenotransplantation. Sodium hydroxide and calcium hydroxide are two common base agents to achieve decellularization. Bases can effectively destroy collagen fibrils and clear various nuclear residues and cellular components of the ECM. Therefore, compared with other chemical reagents and enzyme preparations, the mechanical properties of the ECM are significantly reduced [91]. Ionic or non-ionic detergents, including Triton X-100 and sodium dodecyl sulfate (SDS), can directly destroy cell membranes and eliminate various nuclear residues and proteins from the ECM during the process of decellularization [92]. Currently, many researchers utilize SDS to decellularize many tissues or organs, such as the liver, heart, lung, and kidney [93,94]. However, SDS may also decrease or disrupt the inner structure, reduce the GAG content and essential growth factors, and cause the loss of collagen integrity, leading to alterations in the ECM’s properties [95,96]. In addition, the SDS substance remains in decellularized tissues and might cause inflammation after implantation in vivo. Recently, Kanda et al. utilized liquefied dimethyl ether (DME) as a substitute for SDS to remove lipids from target tissues [97]. They decellularized porcine aortas by combing DME with DNase. The results demonstrated that the decellularization method of DME and DNase could effectively enhance the maximum stress of porcine aortas [97]. Ionic detergents can eliminate the interactions between proteins to achieve decellularization, while non-ionic detergents can clear lipid–lipid and lipid–protein interactions to achieve decellularization. Compared with ionic detergents, non-ionic detergents for decellularization can better protect the inner structure of the ECM [98]. However, non-ionic detergents may be inadequate to decellularize thick tissues, regardless of whether they are combined with SDS [99,100]. However, after decellularization, the residual chemicals in the ECM must be rinsed carefully, especially detergents that penetrate into thick or dense tissues [101]. Moreover, thin tissues (e.g., leaflets) also need to be stirred and cleaned a few times (more than six times) to remove detergents thoroughly [102]. 

### 4.2. Physical Methods

Currently, many researchers use various physical methods, such as freezing and thawing, osmosis, and ultrasound, for the decellularization process. During the process of physical decellularization, physical methods release various components within the cell by disrupting the cell membrane, which are finally removed in conjunction with rinsing [103]. Rapid freezing or snap freezing can achieve decellularization by forming intracellular ice crystals, disrupting cell membranes, and releasing intracellular components. Currently, many researchers utilize rapid freezing or snap freezing to decellularize nerves and tendons [104]. The ECM’s density, tissue thickness, degree of cellularity, and other key factors determine the efficiency of its decellularization [25]. However, the change in temperature rate should be controlled carefully to avoid damaging the ECM’s ultrastructure caused by ice formation [105,106]. Moreover, the membranous and intracellular residues should be further treated to eliminate them. Another decellularization method is nonthermal irreversible electroporation (NTIRE), which mainly uses subtle electrical impulses to cause changes in the potential on the cell membrane, thereby promoting the formation of micropores on the surface of the cell membrane, releasing intracellular components, and achieving decellularization [107,108]. NTIRE can maintain the integrity of ECM networks very well, but it is currently mainly used in the decellularization process of tissues or organs with a small size [25]. Currently, many researchers have successfully used supercritical carbon dioxide (SC-CO_2_) as an alternative to the traditional decellularization methods [109]. The SC-CO_2_ method has advantages of short processing time, nontoxicity, nonflammability, and sterilization effect [110,111]. The SC-CO_2_ technique uses a critical CO_2_ pressure greater than 7.4 MPa to dissolve cells in tissue [112]. Many tissues, such as skin, cartilage, arteries, and corneas, have been fabricated into decellularized tissue substitutes by SC-CO_2_ [113,114,115]. Sawada K et al. reported that a mixture of SC-CO_2_ and ethanol could be utilized to remove the cell nucleus and cell membrane from various tissues or organs under the conditions of 15 MPa and 37 °C [116]. In addition, high hydrostatic pressure (HHP), with a pressure higher than 600 MPa, can directly destroy cell membranes and release various cellular components within tissues [117]. Hashimoto Y et al. reported that HHP can be used to decellularize the porcine cornea in order to apply a corneal scaffold for tissue regeneration [118]. These two techniques can decellularize effectively and preserve the integrity of the ECM network and retain the GAG content, but a medium containing DNase is commonly needed to remove the nuclear residues [118,119]. Physical methods of decellularization have the major advantage of preserving the ultrastructure and ECM components under suitable conditions. Moreover, compared with other decellularization methods, specific equipment such as containers that can withstand high pressure is required, which may not be available in each laboratory [120].

### 4.3. Biological Methods

Biological methods mainly utilize various enzymes and non-enzymatic agents to decellularize tissue or organs. The enzymes used for decellularization mainly include proteases, nucleases, collagenase, dispase, and α-galactosidaseα-galactosidase. Proteases can be used to decellularize tissue or organs by breaking cell–matrix adhesions [121]. The enzyme can remove cell residues with high specificity and destroy cells effectively, while it is difficult to achieve complete cell component removal by utilizing enzymes. The decellularization mechanism of trypsin occurs mainly by breaking the peptide bond between lysine and arginine [122]. Compared with detergents, trypsin, when used in the decellularization process, can directly destroy the ultrastructures in tissues or organs, thereby promoting the infiltration of subsequent decellularization agents and accelerating the removal of various components of cells [123]. Yang B et al. treated the bladder with a mixture of trypsin and ethylenediamine tetra-acetic acid (EDTA) and then submerged it in a hypotonic buffer, Triton X-100, and used nucleases to remove intracellular components to achieve the decellularization of the bladder [124]. This decellularization method can eliminate cellular materials and retain bioactive factors. However, collagens, elastin, and other ECM proteins may resist trypsin cleavage or disruption due to the changes in mechanical properties [25]. Non-enzymatic agents can be used to decellularize tissue as well. Chelating agents, such as EDTA, can directly destroy cell adhesion and can effectively enhance the decellularization effects of other decellularization agents. Researchers demonstrated that the combination of EDTA with other detergents could effectively improve the decellularization effects of detergents but also increased the risk of disrupting the integrity of the ECM network [125]. In addition, toxins such as latronkulin, which is a naturally occurring cytotoxic drug, can also be used for decellularization. Gillies et al. demonstrated that the combination of latrunculin B, hyper- and hypotonic solutions, and DNase could effectively eliminate the cellular components from tissues or organs with a high density [126]. Compared with enzyme and detergent decellularization methods, this method has advantages in removing DNA and retaining GAG. In addition, the mechanical properties of the dECM decellularized by this method are similar to those of native tissue [126]. 

## 5. The Construction and Modification of Decellularized Extracellular Matrix-Based Bioinks

There are three main steps to constructing dECM-based bioinks: initial tissue treatment, tissue decellularization, and post-decellularization processes. Before decellularization, the first step in processing the initial tissue is eliminating connective tissues, large vessels, fat, and other excess impurities. Then, the processed tissue will be cut into small chunks or thin pieces and rinsed with water to remove the residual blood. After preliminary treatment, the next step is decellularization, which includes the elimination of the native cells and protection of the inner structure of the ECM and bioactive components. It is crucial to select the appropriate decellularization methods according to the characteristics of the tissue, which will influence the composition and proportions of the ECM. Each of the three major types of decellularization methods has been discussed above. The last step is the post-decellularization process, which begins with removing the remaining decellularization agents, which are cytotoxic [127]. Then, it is necessary to examine whether the cellular components were removed from the dECM [63]. In addition, compositional analysis is another important aspect for the further application of the dECM [63]. Figure 5 shows the usual methods of confirmation of cell removal and compositional analysis. After this, it is critical that sterilization is performed to remove the pathogenic compounds and avoid an adverse immune response. Various sterilization methods, such as chemical treatments (e.g., ethylene oxide, peracetic acid), dry heat, electron beam irradiation, pressurized steaming, and gamma-ray irradiation, can be employed depending on the biochemical and physical properties of the dECM [128]. Finally, we use instruments and a pestle to lyophilize and pulverize the tissue into small particles. Then, pepsin can be applied to digest and solubilize the dECM powders in an acidic environment. Pepsin-treated dECM solutions can be converted into a gel state at 37 °C, and this method has been used for surface coatings of biomaterials and various hydrogel constructions [129]. 

The dECM is a complicated complex of multiple proteins and polysaccharides, which can regulate cellular functions, including proliferation, anchorage, migration, and signaling, through its surface proteins, cytokine activity, elasticity, and so on, so that the dECM from different tissues possess specific effects for cells [130]. For example, Pei et al. found that the dECM from different types of stem cells consisted of specific proteins and exhibited specific stiffness, which caused the dECM to have varying chondrogenic capacities for synovium-derived stem cells [131]. Beachley et al. further proved that dECM proteins were associated with specific cellular functions, such as the type XII and II collagen associated with osteogenesis and chondrogenesis, and the dECM from some specific tissues contained more specific proteins, which induced the tissue specificity of various dECM derived from different tissues or organs [132]. However, the native dECM derived from different tissues and organs was composed of various proteins; many proteins, and their functions, could not be identified completely, which made the structure of the native dECM very intricate. Thus, to explicate and strengthen the function of the dECM, such as regulating cellular functions and promoting regeneration, the modification of dECM bioinks has become an essential part of 3D bioprinting.

There are several modifications that could be chosen for the dECM before and after bioprinting, including chemical and biological crosslinking; these methods could enhance the bioactivity and mechanical properties of the final scaffolds. Shin et al. modified the cardiac dECM with laponite-XLG nanoclay and poly(ethylene glycol)-diacrylate (PEG-DA) [133]. The results demonstrated that the final bioinks possessed good extrudability, shape fidelity, rapid crosslinking, and cytocompatibility. In addition, its compressive modulus was tunable by changing the proportion of PEG-DA according to the different states of cardiac tissue, which allowed the final scaffolds to approximate the mechanical properties of the native tissue [133]. In another study, Rueda-Gensini et al. fabricated a new dECM-based bioink by mixing graphene oxide (GO) into a methacryloyl-modified decellularized small intestine submucosa (SISMA) hydrogel [134] (Figure 6A). The results demonstrated that methacryloyl biochemical modification could significantly increase the mechanical properties of the dECM-based bioink during the process of extrusion-based 3D bioprinting (Figure 6B). The stabilization of the structure was also an important factor for the bioprinted scaffolds, which influenced the shape fidelity and the activity of cells encapsulated in the bioink [135]. Through modification, the stabilization of the dECM bioink could also be improved. Aromatic rings with three hydroxyl groups could crosslink with proteins in the dECM, causing the bioinks to display more characteristics such as quickly gelatinizing, spontaneous stabilization, shear thinning, and being directly printable on tissues [136]. Moreover, the viscosity of bioinks would affect the bioprinting speed and the final resolution; bioinks with low viscosity would lead to the deformation of the printed scaffolds and affect the final resolution [137]. In addition, high-viscosity bioinks would increase the extrusion pressure, harming the viability of cells encapsulated in the bioink [138]. Sobreiro-Almeida et al. [139] lyophilized and digested porcine kidneys to regulate the viscosity of the solution and used agarose for the enzymatic crosslinking of the dECM-based bioink. Moreover, alginate was also suggested as a printable hydrogel that could be mixed with the ECM and increase the viscosity of bioinks [140]. For the application of 3D bioprinting, via these modifications of the dECM, we could produce scaffolds with appropriate mechanical strength, a stable structure, and tissue-specific microenvironmental niches for cell proliferation and migration.

## 6. The Applications of Decellularized Extracellular Matrix-Based Bioink for Bioprinting in Tissue Engineering

The tissue-specific biochemical characteristics of the dECM derived from various tissues or organs make it an attractive option for the fabrication of bioprinted tissue or organs. Recently, growing interest has been focused on biomimetic tissue constructs from dECM-based bioinks, which could recreate the native tissue’s structure, function, and content when employed for tissue or organ regeneration, based on the bioactive effect of the dECM in tissue remodeling [141]. To date, a variety of 3D bioprinting applications have used dECM-based bioinks for a variety of tissue regeneration processes. Table 1 shows the recent applications of dECM-based bioinks for bioprinting in tissue engineering. In recent years, more and more researchers have utilized the dECM as a bioink to fabricate various tissue or organ substitutes by 3D bioprinting. In this section, we enumerate the applications of various dECM-based bioinks in tissue repair and regeneration by 3D bioprinting. 

### 6.1. Hearts

The heart is an intricate organ that has complex structures and regular ejection, so the scaffolds used in cardiac regeneration should have strong mechanical properties and complex structures. The earliest 3D bioprinting of a dECM derived from heart tissue (hdECM) was reported by Pati et al. [142], which was printable without a supporting framework (Figure 7). Subsequently, the study confirmed that the scaffold based on hdECM had positive effects on the functional maturation of myoblasts by analyzing the expression levels of genes of the fast myosin heavy chain [142]. Although the mechanical properties of the first 3D bioprinting constructs from dECM-based bioink did not meet the application demands of cardiac tissue, the study revealed the superiority of hdECM bioinks, which could reconstruct a more favorable native microenvironment on the culture of encapsulated cells compared with that of the collagen bioink. Based on these results, a prevascularized stem cell patch was developed for cardiac repair and used in the rat myocardial infarction model to improve cardiac function [30]. The outcomes indicated that the microenvironment in the patch could enhance neovascularization and cardiac tissue regeneration, which demonstrated the suitability of the bioprinted patch for cardiac tissue repair and regeneration [30]. In order to improve the mechanical performance of bioprinted scaffolds constructed by dECM-based bioinks, vitamin B2 was added to dECM bioinks via UVA irradiation [143]. These bioinks had high fidelity and similar biomechanical properties to native cardiac tissue so that the scaffolds could mimic the native microenvironment of cells [143]. Moreover, gelatin methacrylate was also added in hdECM bioinks to enhance the mechanical characteristics of the dECM-based scaffolds constructed by 3D bioprinting [144]. Notably, the UV exposure time after bioprinting can also be changed to regulate the mechanical characteristics of dECM-based scaffolds [144]. Moreover, Das et al. [145] proposed that external stimuli could affect cardiac regeneration in bioprinted scaffolds. They printed hdECM bioinks in an extruded polyethylene vinyl acetate (PEVA) construction and stimulated the encapsulated cardiomyocytes via applying stretch stimuli outside the hdECM-PEVA construct [145]. Under the dynamic stimulus, cardiomyocytes showed enhanced maturation with high expression of sarcomeric patterns, synthesizing cardiac troponin T proteins and intracellular calcium transience [145]. The results demonstrated that the native microenvironment of dECM-derived bioinks can act as the decisive factor to regulate cardiomyocyte maturation. In addition, the interactive mechanisms of the cell matrix under the microenvironment still need further research.

### 6.2. Cartilage

Due to its unique mechanical characteristics, high water content, and lack of blood vessels, the repair and regeneration of cartilage tissue represent a continuing challenge in the field of tissue engineering and regenerative medicine [168]. In recent years, the development of 3D bioprinting technology and material sciences has brought fresh opportunities for the repair and regeneration of cartilage tissue [169]. Cartilage-derived dECM (cdECM) bioinks have great potential as biological inks to construct cartilage scaffolds. They are derived from autologous tissue and have a good ability to regulate cellular behaviors, such as proliferation and migration. In addition, the cdECM can also direct the cellular differentiation of stem cells encapsulated in bioinks. Initially, Pati et al. [142] demonstrated that cdECM bioinks were printable, with the polycaprolactone (PCL) framework supporting them. However, the ECM of cartilage is relatively denser than that of other connective tissue, causing the decellularization to require further treatment with enzymes, which leads to a reduction in COL and GAGs in the cdECM [142]. In addition, the in vitro results demonstrated that the microenvironments of the cdECM could effectively direct mesenchymal stem cells to differentiate into chondrocytes [142]. Jung et al. [146] developed a dECM–silk bioink by physically crosslinking a cdECM and silk fibroin; this type of bioink had controllable viscosity and better tissue differentiation. Zhang et al. [147] fabricated a crosslinker-free bioink also containing silk fibroin and cdECM; the final scaffolds had a porous structure, good mechanical properties, and suitable degradation performance. Furthermore, it was proven that the cdECM-based scaffolds could promote the chondrogenic differentiation of bone marrow mesenchymal stem cells (BMSCs) and significantly increase the chondrogenesis-specific genes’ expression [147]. Chae et al. [148] fabricated a bioprinted meniscus construct containing polyurethane, PCL polymers, and meniscal dECM. The bioinks had high controllability and durable architectural integrity, so the printed construct had excellent mechanical properties and tensile properties to resist external pressure [148]. Lu et al. have successfully developed a dECM-based bioink, by mixing dECM and poly (vinyl alcohol) (PVA), for meniscus repair and regeneration [149]. The in vitro results demonstrated that the bioprinted scaffold possesses an excellent deformation capability (Figure 8). Although these 3D-printed constructs have not completely realized a realistic function and structure, they verify the superiority of cdECM bioinks and 3D printing technology in cartilage regeneration.

### 6.3. Adipose Tissue

A potentially effective strategy in treating soft tissue defects is adipose tissue engineering. Therein, decellularized extracellular matrix (dECM) scaffolds with excellent adipogenic induction capabilities show promise in improving soft tissue. Firstly, Pati et al. [142] successfully developed a dECM derived from an adipose tissue (adECM) bioink and printed it throughout a PCL support to fabricate tissue constructs. Through the decellularization process, the level of collagen slightly increased and the level of GAGs moderately decreased in the adECM [142]. Via immunofluorescence staining, adipogenic markers in mesenchymal stem cells, such as PPARγ and LPL, were significantly increased in the adECM group [142]. Then, in their following research, Pati et al. [170] successfully utilized adECM and PCL to fabricate a dome-shaped bioprinted dECM-based scaffold, which could promote the adipogenic differentiation of adipose-derived mesenchymal stem cells (ADSCs). Then, the researchers utilized the adECM-based bioink to encapsulate ADSCs and implant them into the subcutaneous tissue of mice. The in vivo staining results presented a potent angiogenetic response around the construct and new adipose tissue formation [170]. Moreover, the study proposed that the cells at the center layer would generate hypoxia after 14 days of culture, with cell viability significantly decreased, which should be overcome in future research [170]. Currently, the low cell density of adipose tissue substitutes often leads to immature adipogenesis. Fabricating packed adipogenic lipid droplets is helpful in rebuilding the morphology of native adipose tissue. In order to recreate the morphology and physiological functions of adipose tissue, Ahn et al. utilized a hybrid bioink, composed of adipose dECM and alginate, to fabricate fully mature, densely packed adipose tissue by environmentally controlled in-bath 3D bioprinting technology [150]. The results confirmed that the human subcutaneous preadipocyte cells encapsulated in the hybrid bioink could differentiate into lipid-accumulating mature adipocytes [150]. In another study, Amo et al. combined decellularized adipose tissue with plasma and fibroblasts to fabricate a novel composite bioink [151]. The results demonstrated that the composite bioink could provide a suitable microenvironment to regulate the cellular behaviors of fibroblasts. Adipose tissue transplantation can also be used to encapsulate various glands and maintain their biological functions [171,172]. Yu et al. minced parathyroid glands into fragments with small sizes and encapsulated them into adECM-derived bioinks for bioprinting [152]. The results demonstrated that the adECM-derived bioinks could preserve the biological function of the parathyroid glands in vivo [152]. Currently, there are still few studies focusing on applying adECM in bioprinting. In addition, the issue of how to optimize the physicochemical properties of adECM for better application in bioprinting still needs further research.

### 6.4. Skeletal Muscle

Skeletal muscle has a unique structural characteristic, being composed of several bundles of fibers that are perfectly aligned. Although skeletal muscle has a remarkable capacity for self-healing, trauma-induced permanent volumetric and functional loss may result in functional impairment, disability, and chronic pain [173]. Recently, growing interest has been focused on 3D printing technology and muscle dECM (mdECM) bioinks for skeletal muscle regeneration. Due to the constant contraction and expansion of muscle, the construct should possess realistic mechanical properties and an anisotropic microenvironment. Choi et al. [142,174] developed an mdECM bioink, which preserved the complex bioactive components, including GAGs, collagen, and other bioactive agents. The bioink presented favorable printability and promoted myogenic specification, enabling the 3D-printed structures to produce observable contraction in response to electric stimulation. These features suggested that, prior to implantation, the mdECM bioink may be utilized to design and create a real architecture of injured muscle mass and the constructs could provide cells with a realistic-like microenvironment, facilitating tissue development and maturation [174]. In order to enhance the mechanical properties, structural integrity, and shape fidelity, a granule-based printing reservoir was created by Choi et al. [153] to enable the construction of functional volumetric muscle constructs utilizing soft dECM bioinks without changing the chemical components and structural fidelity. This technology allowed the construct to show higher mechanical properties and an organized microenvironment, with potential to use the mdECM bioink to produce the inner core and using the dECM derived from vascular tissue to produce the outer shell [153]. In another study, Kim et al. [154] have successfully fabricated a novel bioprinting method, using crosslinked mdECM and methacrylate (mdECM–MA), and combined it with fibrillated poly(vinyl alcohol) (PVA), that could biochemically and topographically mimic skeletal muscle constructs. The mdECM–MA bioink was a photo-crosslinkable material, which preserved the collagen, GAGs, elastin content, and growth factors in the dECM so that the bioink was able to promote the proliferation and myogenesis of myocytes [154]. Following the addition of PVA fibrils, the printed constructs presented unique topological cues, which would induce cellular alignment and enhance myogenic differentiation [154]. 

### 6.5. Liver

The use of liver dECM (ldECM) bioinks in 3D-printed liver constructions has been extensively studied in hepatic regeneration research. A major difficulty in these studies is preserving and improving hepatocyte phenotypes and functions. Initially, Skardal et al. [175] developed a type of bioink with ldECM, hyaluronic acid (HA), and gelatin via the multi-crosslinking method. The first step was the crosslinking of thiol-acrylate, which formed a soft and extrudable material for bioprinting; then, the thiol-alkyne polymerization reaction was initiated by UV light to stabilize the final constructs. The stiffness of the bioink spanned from 113.66 Pa to 19.798 kPa, with potential to mimic all soft tissue in the body [175]. The constructs, printed by primary hepatic cell and ldECM bioinks, presented structural stabilization, high cell viability, albumin production, and urea secretion [175]. Lee et al. [155] developed a type of printable ldECM bioink that mainly comprised GAGs, fibronectin, and collagen. The biochemical performance of the dECM-based bioinks can be regulated by altering the dECM concentration. The researchers finally chose the 3% ldECM bioink, which presented a similar modulus to native liver tissue [155]. The bioink showed the expected printability and high cell viability for BMSCs and human hepatocellular carcinoma (HepG2) cells. Moreover, the ldECM bioink showed the facilitation of BMSC differentiation and was able to promote HepG2 cells to secrete albumin and urea [155]. By combining ldECM bioinks with digital light processing (DLP)-based 3D bioprinting technology, Yu et al. [144] hypothesized that it would be possible to quickly tune the mechanical properties while also creating intricate, high-resolution microscale geometries and promoting human-induced pluripotent stem cells (iPSCs) to differentiate into hepatocytes. Mao et al. [156] also used DLP technology in bioprinting liver constructs. They developed a type of bioink combining photocurable methacrylate gelatin (GelMA) with ldECM and fabricated liver microtissues. The human-induced hepatocytes encapsulated in the microtissues showed high viability and better functions. Ma et al. [156] also utilized DLP technology and a dECM-based bioink, consisting of GelMA and dECM derived from the liver, to fabricate bioprinted liver scaffolds with different stiffness performance to mimic cirrhotic liver tissue, and they investigated the cellular behavior of HepG2 cells in disease modeling.

### 6.6. Skin

The skin is the first line of the body’s defense against injuries from the outside world. Thus, skin defects are one of the most common issues in the clinical setting. As skin tissue has complex structures and functions, it is difficult to investigate the cellular behavior and mimic native structures via 2D cell culturing. Recently, 3D bioprinting technology has attracted more attention in producing full-thickness skin models that could be used for skin regeneration engineering, cosmetics testing, and the investigation of cellular function [176,177]. Kim et al. [157] developed a printable bioink from porcine skin-derived dECM (sdECM) and investigated its biochemical performance through in vitro and in vivo evaluations. The results of in vivo experiments demonstrated that the sdECM 3D-printed skin had more physiologically relevant barrier properties, a stable dermal compartment, dermal secretion, and superior epidermal organization [157]. In in vitro research, the sdECM-based 3D skin patch showed remarkably stimulated re-epithelialization, neovascularization, and wound closure in mouse dorsal wound modeling [157]. Won et al. [158] developed a form of bioink combining porcine skin-derived dECM and human dermal fibroblasts (HDFs). The results demonstrated that cell viability and the expression level of skin morphology-related genes in the bioprinted constructs was increased in HDFs [158]. Ahn et al. [159] have successfully developed a novel bioprinting technique using bioprinting equipment installed with heating modules to control the thermal crosslinking of the constructs during the process of bioprinting in order to evaluate the printed bioink gels simultaneously and analyze the effect of the crosslinking extent on printability. Moreover, Kim et al. [178] also developed a novel bioprinting technique to fabricate a fully matured 3D skin model with a perfusable and vascularized epidermis, dermis, and hypodermis. The fabrication of a skin-derived collagen bioink and the 3D stacking process are shown in Figure 9. The skin model presented similar maturation to native human skin through the high expression levels of differentiation markers of the epidermis, including keratin 10, involucrin, and filaggrin [178]. This technology has the potential to open up pathological research pathways and simulate skin diseases. In another study, Kim et al. utilized bioprinting technology to fabricate a diseased skin tissue model with pathophysiological hallmarks of type 2 diabetes by using a skin-derived dECM [179]. Differing from other studies focusing on the development of diabetic animal models, this study utilized 3D diseased skin tissue to fabricate engineered human skin models to investigate the pathophysiology of the skin response [179]. The use of a dECM derived from diseased tissue can fully mimic the pathological microenvironment, which facilitates the in vitro study of cell–pathogenic molecular interaction mechanisms in disease. Although many 3D skin bioprinting techniques have been announced in recent years, there are still many difficulties to overcome before the bioprinting technique can successfully regenerate the skin function clinically. Inadequate vascularization methods for bioprinted skin substitutes, a lack of biomaterials with both promising printability and compatibility, and the challenge of clarifying how skin appendages interact with nerves and veins are still challenging when bioprinting skin substitutes.

### 6.7. Cornea

In corneal tissue engineering, the fundamental function of the artificial cornea is to replicate the rotational symmetric curvature, which is necessary for optical refractive power [180,181]. In previous studies, a corneal dECM-based hydrogel or sponge had been used for corneal repair, but it was difficult to maintain its transparency [182]. Recently, 3D-printed corneal constructs have been used in the characterization of corneal cellular regeneration and modeling for corneal fibrosis [183]. However, the optics remain one of the most significant difficulties because the smooth surface needs a strong solution from the bioink; moreover, the production process involves bioprinting flat layers into a curved structure [184]. The native cornea has dense collagen fibrillar structures; the cornea-derived dECM mainly comprises collagen and is able to maintain the keratocyte phenotype [160]. Park et al. decellularized the corneal ECM containing differentiated corneal stromal cells to fabricate a 3D-bioprinted decellularized collagen sheet (3D-BDCS), and they utilized SS-OCT to conduct the in vivo non-invasive monitoring of 3D-BDCS after implantation onto rabbit corneas [185] (Figure 10A,B). Figure 10C shows the slit-lamp microscopic images after implantation onto rabbit corneas, demonstrating that 3D-BDCS exhibited good biocompatibility in vivo. Kim et al. [161] have successfully fabricated a type of printable bioink based on the corneal dECM to fabricate functional corneal constructs. In in vitro assessments, the corneal constructs showed optical transparence and similar biochemical performance to the native cornea. The turbinate-derived MSCs encapsulated in the bioinks displayed a higher expression level of keratocan after 14 days of culture [161]. These results revealed that the corneal dECM bioink could offer a more favorable microenvironment for corneal regeneration.

### 6.8. Brain

Currently, the studies of brain-derived dECM bioinks are in the preliminary stages. Yi et al. [162] successfully fabricated a dECM-based bioink derived from the porcine brain to bioprint patient-specific glioblastoma (GBM)-on-a-chip, which was used to identify patients’ responses to chemoradiotherapy. They fabricated reconstituted glioblastoma tumors via the 3D bioprinting method, which comprised tumor cells, a dECM derived from the porcine brain, and endothelial cells. In the brain-derived dECM-based bioink, the expression of genes related to ECM remodeling and encoding pro-angiogenic factors was higher than that in a collagen dECM bioink. Moreover, the GBM-28 cells presented higher invasiveness and a more spindle-like morphology in the brain-derived dECM bioink [162]. Based on the 3D bioprinting technology, they engineered a patient-specific GBM chip, which could capture the key properties of the tumor environment, including the microenvironment in the constructs and the compartmentalized structures surrounded by the vascularized stroma and the oxygen gradient [162]. In addition, immunofluorescence demonstrated that the brain dECM bioink might offer similar pathological characteristics, including an oxygen gradient that causes hypoxia and the dysfunctional microvessels surrounding the tumor [162]. Through evaluating the treatment resistance shown by the GBM cells, the GBM chip appeared more useful in the prediction of the effectiveness of clinical therapies [162]. Collectively, this GBM chip model has favorable potential in identifying the effects of treatment in GBM patients.

### 6.9. Pancreas

The pancreas is responsible for regulating blood sugar and secreting digestive enzymes. Following the increase in type 1 diabetes mellitus (T1DM), the transplanted islets produced by tissue engineering have become a significant opportunity to improve pancreatic function [186]. In the pancreas, the interactions between islets and the ECM play crucial roles in regulating pancreatic β-cell function [187]. Therefore, the pancreatic dECM could provide a critical microenvironment for tissue regeneration. Kim et al. [163,164] developed pancreatic dECM bioinks and investigated their viability, insulin secretion, and glucose responsiveness. In the in vitro assessments, the pancreatic dECM bioink showed favorable viability for islets, and in the glucose-induced insulin secretion test, the islets encapsulated in pancreatic dECM bioink presented high sensitivity to glucose, secreting more insulin. When the islets were co-cultured with human umbilical vein endothelial cells (HUVECs) in the pancreatic dECM bioink, the vascularization and interactions between cells were enhanced, which improved the efficacy of the islets. Moreover, the hiPSC-derived insulin-producing cells showed a high expression level of the pancreatic and duodenal homeobox 1 (PDX1) gene, insulin (INS) gene, and glucagon (GCG) gene. Collectively, these results confirmed that the pancreatic dECM bioink could provide a suitable microenvironment for islets and HUVECs, and the modeling of the combination of islet and HUVECs cultured in a pancreatic dECM bioink could be used in further research on pancreatic regeneration. 

### 6.10. Trachea

The airway has a complex microenvironment due to its daily exposure to environmental agents. Park et al. [165] utilized a porcine-derived tracheal mucosa (tmdECM) bioink within a PCL support to develop a type of functional airway-on-a-chip. The tmdECM showed a variety of angiogenic factors and promoted angiogenic differentiation, forming a stable vascular network. Moreover, through the tmdECM bioink combined with endothelial cells and fibroblasts, the study fabricated a functional interface between the airway epithelium and vascular network. Furthermore, Park et al. [166] developed a tracheal mucosa (tmdECM)-based bioink and fabricated a tracheal scaffold with PCL support. They used the tmdECM scaffolds ladened with human inferior turbinate mesenchymal stromal cells (hTMSCs) to repair tracheal defects in the rabbit model. The bronchoscopy examinations showed that the tmdECM scaffolds were completely covered by epithelial tissue two months after the operation, while the collagen scaffold with hTMSCs group showed serious stenosis. Immunofluorescence showed the expression of KRT-14 in the basal cells, which revealed the active regeneration of basal cells. These results confirmed that the bioprinted tracheal graft is an appropriate tissue engineering strategy for extensive circumferential tracheal reconstruction. It is possible to fabricate high-function organ models that mimic complex organ environments through 3D bioprinting in the future. 

### 6.11. Blood Vessels

In tissue engineering, blood vessels in the tissue and organs could provide oxygen and various metabolic nutrients to repair and regenerate tissue. When the cells or tissue lack a blood supply, they could die from hypoxia [188]. Thus, it is a crucial step to fabricate vasculature for any tissue-engineered constructs. Gao et al. [31] developed a hybrid bioink composed of vascular-tissue-derived dECM (VdECM) and alginate; they fabricated bio-blood vessels by combining hybrid bioinks, endothelial progenitor cells, and PLGA microspheres. In mouse models with ischemic disease, the transplantation of bio-blood vessels could promote neovascularization in ischemic limbs and recover the ischemic limbs. This is also the first demonstration of the use of a dECM hydrogel bioink as a drug carrier during bioprinting. Based on these results, Gao et al. [167] constructed a freestanding in vitro vascular model, including the bioprinting of vascular equivalents and a perfusion platform, which could print a complex construct of vessels with complete tunable perfusions. The immunofluorescence result proved that the vessel constructs formed an intact endothelium and tight junction after seven days of culture without additional cell seeding [167]. Moreover, the vessel constructs showed the ability to sprout of neovessels in response to the stimulus of proangiogenic factors, which laid a foundation for the development of advanced models for observing the process of angiogenesis. Xu et al. [189] developed a cartilage-derived dECM bioink, combined with Pluronic F127, used to mimic native vessels. Through 3D bioprinting technology, they fabricated thick tissues with multilevel vascular structures, which were composed of three layers, including intima laden with HUVECs, a media layer laden with human aortic vascular smooth muscle cells (HA-VSMCs) and dECM, and adventitia laden with human dermal fibroblasts–neonatal (HDF-n) cells. Meanwhile, Xu et al. suggested that this method could be utilized to build unique structures according to arteries, veins, and the diameters of vessels by changing the composition of each layer of construct [189]. 

### 6.12. Tendon

The tendon, which distributes tension during movement and joins the skeletal muscle to the bone, is composed of dense connective fibrous tissue. The tendon has strong mechanical properties, with ultimate tensile strength ranging from 50 to 150 MPa in the human body [190]. The great tensile strength of the tendon is mainly a result of its ECM components. At present, the mechanical properties of an artificial tendon are difficult to achieve at this level, so the tendon-derived dECM bioink provides a new approach to tendon regeneration. Toprakhisar et al. tried to develop a tendon-derived dECM-based bioink to maintain the native mechanical properties of tendons [191]. They found that the mechanical properties of the dECM were influenced by its concentration and digestion times, and the gelation kinetics were mainly influenced by its concentration. They created 3D structures from a tendon dECM bioink by controlling the temperature and gelatinization time, without using any crosslinkers or supports [191]. The results demonstrated that the mechanical qualities of the 3D structures were lower than those of native tendon tissue. The 3D structures could induce ADSCs to differentiate into tenocyte lineages. Additionally, during maturity, the cells encapsulated in the dECM bioink displayed a well-aligned fiber orientation. These results revealed that the tendon dECM bioink could promote tendon-specific differentiation and enhance the development of tendon-like tissue. 

## 7. Current Challenges and Further Perspectives

### 7.1. Optimizing the Fabrication Procedure of dECM-Derived Bioinks

Compared to other materials, dECM-derived bioinks contain various bioactive components and mimic the cellular microenvironment of native tissue. In addition, dECM-derived bioinks can effectively regulate the various cellular behaviors of seed cells. The composition of the dECM varies across different tissue sources during the process of decellularization. Moreover, reagent residues in dECM-derived bioinks may cause the death of seed cells and the occurrence of immune rejection. Avoiding the use of toxic and difficult-to-remove decellularized substances can enhance the biosafety of dECM-derived bioinks. In addition, developing reasonable detection methods can help to detect the presence of residual toxic substances in dECM-derived bioinks before bioprinting. Meanwhile, the different decellularization processes also cause changes in the components of the dECM [192]. For different tissues, the establishment of a unified decellularization process is conducive to the further clinical application of dECM-based materials. In addition, inappropriate sterilization methods would impair the physical and chemical properties of dECM-based bioinks. Currently, there are few studies focusing on the differences in the sterilization methods of dECM-derived bioinks. Exploring appropriate sterilization protocols for different dECM-derived bioinks is an important aspect in optimizing the fabrication procedure and still needs to be studied in depth.

### 7.2. Mechanical Properties

In the fields of tissue engineering and regenerative medicine, the mechanical properties of the dECM have been a long-standing issue. Currently, most researchers utilize pepsin to digest various tissues, which impairs the mechanical properties of the dECM [193]. During efforts to enhance the printability of bioinks, the decellularization and solubilization processes severely damage the structures and mechanical properties of dECM-derived bioinks. In addition, the cellular behaviors of encapsulated cells can be affected by the mechanical properties of dECM-derived bioinks [194]. Many methods have been used to improve the mechanical properties of dECM-derived bioinks, such as adding supporting materials [142] and chemical crosslinking [156]. Various materials, such as PEG and GelMA, are mixed with the dECM to fabricate hybrid bioinks with better mechanical properties. In addition, many nanoparticles, such as hydroxyapatite, graphene oxide, magnesium oxide, and zinc oxide, can be encapsulated into dECM-derived bioinks to enhance the mechanical properties and bioactivities [195]. However, the issue of how to fabricate larger-scale tissue substitutes with similar mechanical properties to native tissue still needs further research. 

### 7.3. Long-Term Biosafety of dECM-Derived Bioinks after Implantation

Currently, most of the dECM-derived bioinks are derived from porcine tissues, which might increase the potential risk of immune rejection due to species differences. The in vivo biosafety of a xenogeneic dECM after implantation still remains unclear. During the fabrication of dECM-derived bioinks, most potential immunogenic agents affect nucleic acids, cell membranes, and cytoplasmic structures [196]. However, the long-term risk of immune rejection of remaining immunogenic agents in dECM-derived bioinks still remains unclear. Recently, in a large animal trial, a heart-derived dECM was injected into the endocardial region for biosafety evaluation [197]. After injection, there were no deterioration events, such as thromboembolism, inflammation, ischemia, or arrhythmia [197]. Meanwhile, in a small-scale clinical trial, a heart-derived dECM hydrogel was used in 15 patients with moderate left ventricular dysfunction [198]. After implantation, no serious adverse events occurred in short-term follow-ups. Long-term follow-ups of patients treated with dECM-derived bioinks are needed. Due to the hazards associated with transplantation, a more standardized selection of dECM sources should be used. In addition, the standardized quality control of dECM-derived bioinks is required in further research.

### 7.4. Drug Discovery and Development

The development of clinical drugs is inseparable from traditional cell experiments, such as biosafety evaluation [199]. Currently, the 2D monolayer cell culture model is a commonly used in vitro drug screening method. Researchers perform in vitro biosafety evaluation by investigating the effects of drugs on cell behaviors, such as proliferation, differentiation, and migration [200,201]. However, this 2D monolayer cell culture model cannot adequately mimic the microenvironments of various native tissues, resulting in studies that cannot accurately evaluate the tissue’s response to drugs. In addition, animal models are unable to accurately predict the drug effects in humans due to natural differences between species. In recent years, more and more researchers have constructed 3D cell culture models that can simulate natural tissue microenvironments for drug screening [202]. The dECM can adequately mimic the microenvironments of various native tissues and can also induce the directed differentiation of seed cells. Bioprinting technology combined with dECM-derived bioinks can be used to fabricate personalized 3D tissue models, which fully simulate the cell–cell and cell–matrix interactions in native tissues [203]. In addition, the bioprinted tissue models fabricated via the dECM can accurately regulate the distribution of various bioactive factors in the 3D construct [204]. However, there are few studies focusing on the 3D bioprinting of dECM bioinks for drug testing and development. In addition, the differences between bioprinting technologies in the construction of drug screening models, the question of how to construct multicellular drug screening models, the isolation of patient-specific cells, and the printing resolution still need further research.

## 8. Conclusions

As a popular biomaterial derived from native tissues in regenerative medicine, the dECM not only preserves the main ECM components but also possesses low immunogenicity after removing lipids, nucleic acids, membranes, cytoplasmic structures, and other immunogenic substances. The various ECM components in a dECM-based bioink can regulate cellular functions, modulate the tissue regeneration process, and adjust tissue remodeling. In addition, the dECM has been utilized to fabricate tissue constructs for diseased tissue models, drug screening, tissue regeneration, pathology studies, and targeted drug and cell delivery. Due to its outstanding biochemical performance, more and more researchers consider the dECM as one of the best options for organ/tissue substitutes fabricated by bioprinting technology. Additionally, the dECM as a platform might offer a biomimetic milieu for integration with stem cells and other bioactive materials, providing a viable model for the creation of future scaffolds. We believe that, with the rich interdisciplinary research in the fields of engineering, biomaterials science, stem cell biology, and medicine, dECM bioprinting will mature into a viable 3D bioprinting technique for tissue engineering and regenerative medicine.

## Figures and Tables

**Figure 1 materials-16-03197-f001:**
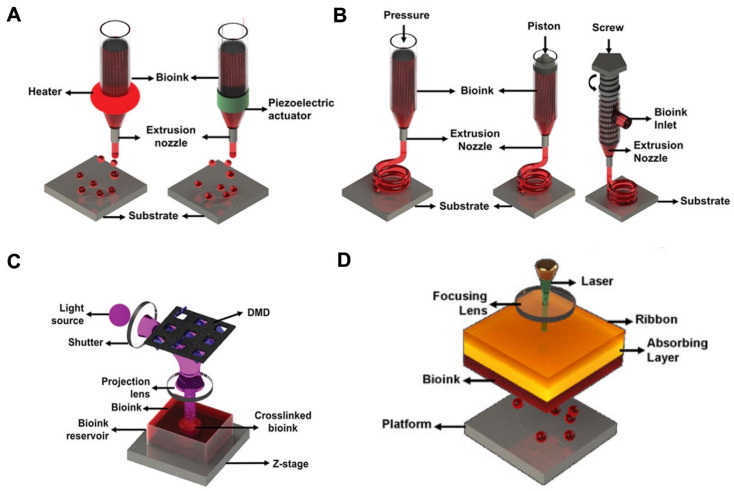
Schematic representation of four bioprinting methods [33]. (**A**) Inkjet-based bioprinting. (**B**) Extrusion-based printing. (**C**) Stereolithography-based bioprinting. (**D**) Laser-assisted bioprinting.

**Figure 2 materials-16-03197-f002:**
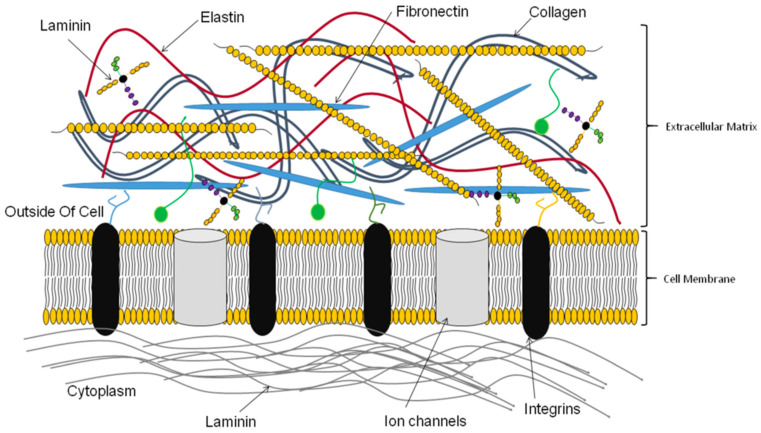
Various ECM components [68].

**Figure 3 materials-16-03197-f003:**
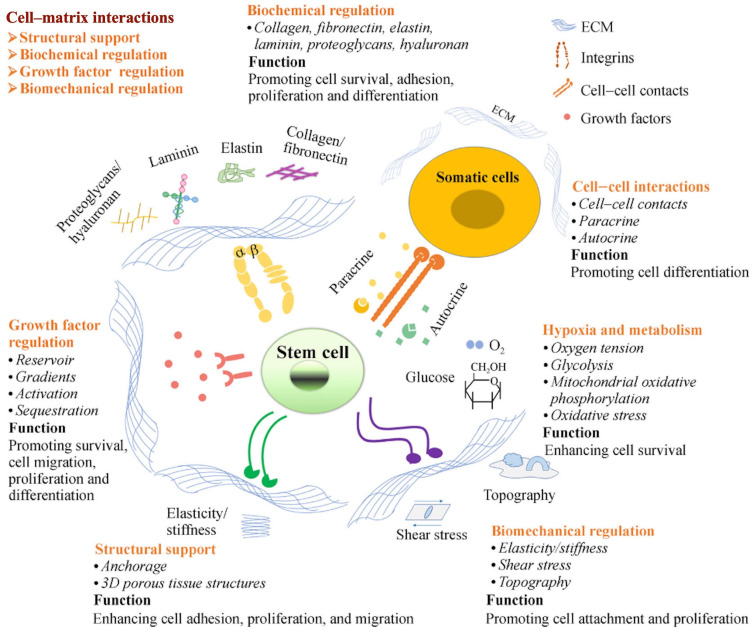
The role of the ECM acting as the cellular niche of stem cells [77].

**Figure 4 materials-16-03197-f004:**
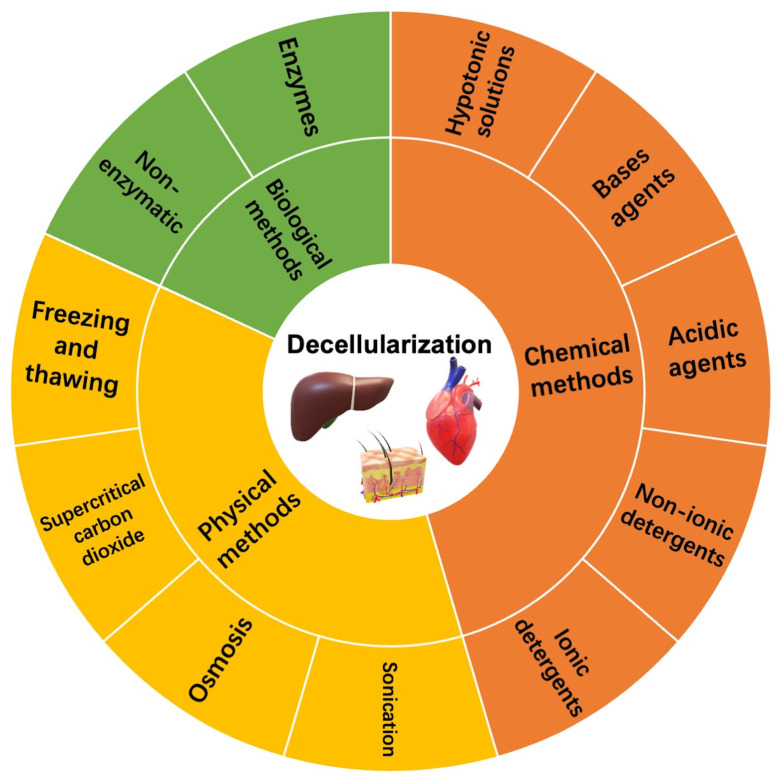
The common decellularization methods.

**Figure 5 materials-16-03197-f005:**
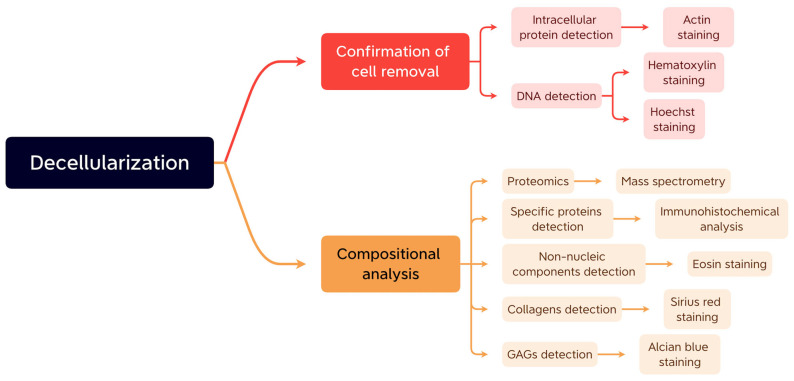
The usual methods of confirmation of cell removal and compositional analysis.

**Figure 6 materials-16-03197-f006:**
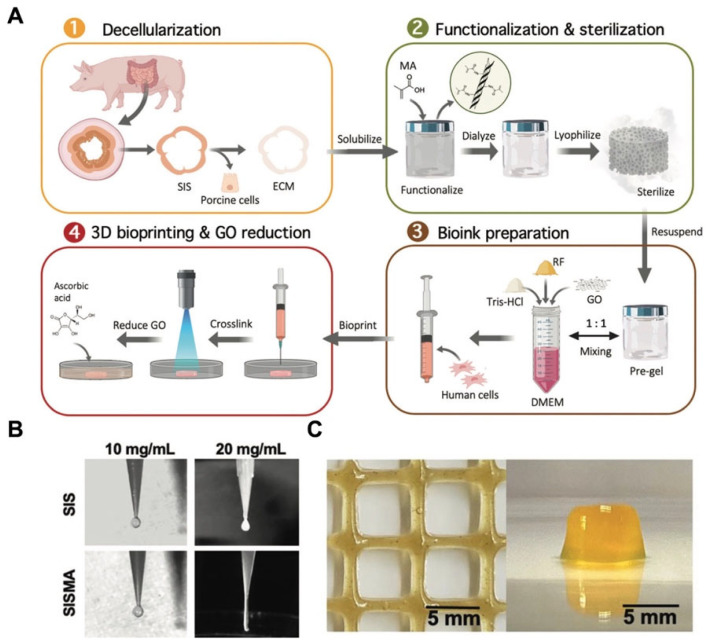
(**A**) A schematic illustration of the fabrication of SISMA bioinks [134]. (**B**) The filament formation of SIS and SISMA during the process of bioprinting. (**C**) The SISMA-GO composite hydrogels [134].

**Figure 7 materials-16-03197-f007:**
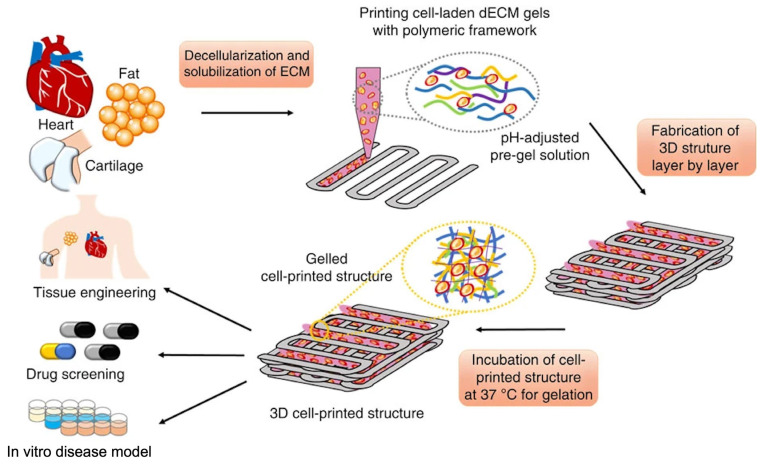
The process of using dECM-based bioink to fabricate dECM-based scaffolds [142].

**Figure 8 materials-16-03197-f008:**
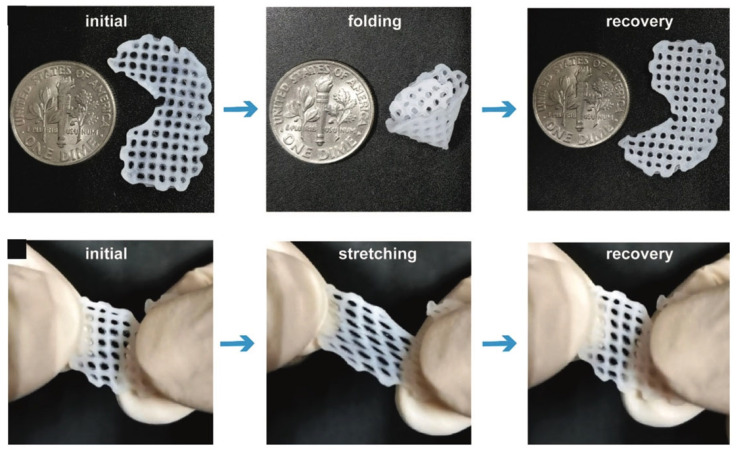
The folding and stretching of a bioprinted scaffold [149].

**Figure 9 materials-16-03197-f009:**
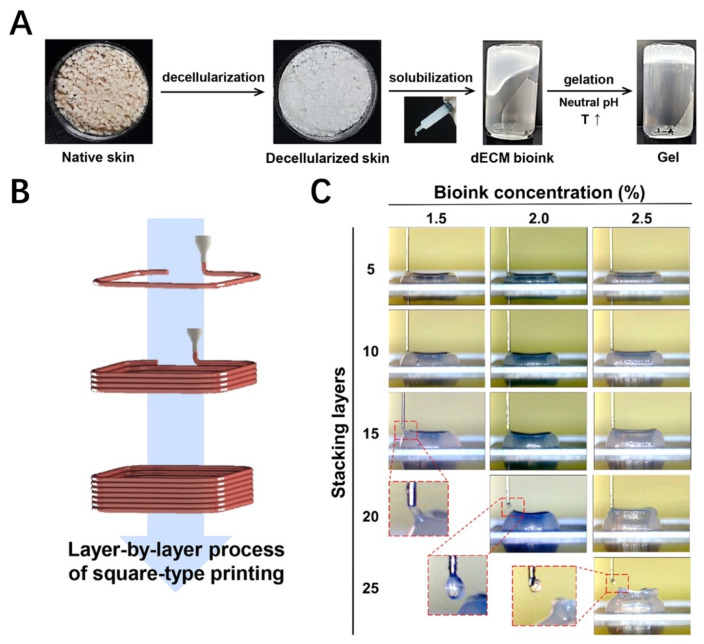
The fabrication of skin-derived collagen bioink and the 3D stacking process [178]. (**A**) The fabrication of skin-derived collagen bioink. (**B**) 3D stacking process. (**C**) 3D stacking with different concentrations.

**Figure 10 materials-16-03197-f010:**
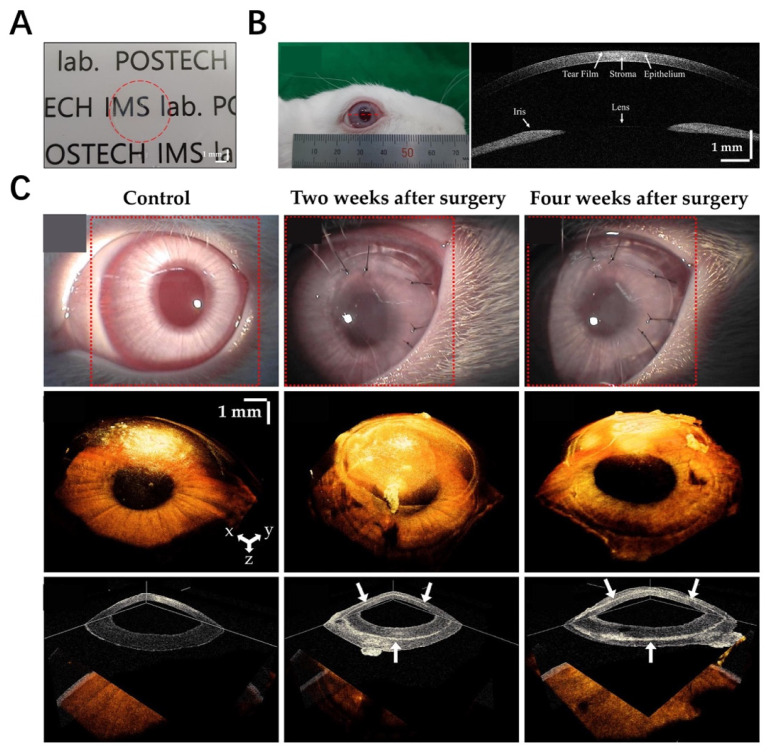
(**A**) 3D-bioprinted decellularized collagen sheet [185]. (**B**) In vivo implantation experiment [185]. (**C**) Slit-lamp microscopic images [185].

**Table 1 materials-16-03197-t001:** The recent applications of dECM-based bioinks for bioprinting in tissue engineering.

Bioink Composition	Seed Cells	Biological Factors	Source of dECM	Applications	References
Decellularized heart tissue	Rat myoblast cells	-	Porcine heart	Heart tissue regeneration	[142]
Decellularized heart tissue	Cardiac progenitor cells	VEGF	Left ventricle from the complete porcine heart	Hydrogel patch for cardiac repair	[30]
Decellularized heart tissue	Human cardiac progenitor cells	Vitamin B2	Heart tissue from a 6-month-old Korean domestic pig	In vitro fabrication of engineered tissue	[143]
Decellularized heart tissue	Human iPSCs	-	Heart left ventricles from Yorkshire pigs	Fabrication of patient-specific tissue model	[144]
Decellularized heart tissue	Neonatal rat cardiomyocytes	-	Heart tissues from 6-month-old Korean domestic pigs	Engineered heart tissue	[145]
Decellularized cartilage tissue	Human inferior turbinate-tissue-derived MSCs	-	Porcine cartilage tissue	Cartilage tissue regeneration	[142]
Decellularized cartilage tissue and silk fibroin	Rabbit bone-marrow-derived MSCs	-	Porcine articular cartilage	Developing tissue substitutes with irregular shape	[146]
Decellularized cartilage tissue and silk fibroin	Rabbit bone-marrow-derived MSCs	TGF-β3	Articular cartilage tissue from female goats	Cartilage regeneration	[147]
Decellularized menisci and polyurethane and polycaprolactone polymers	Human bone-marrow-derived MSC	-	Porcine lateral and medial menisci	Meniscus regeneration	[148]
Decellularized menisci and poly(vinyl alcohol)	-	-	Rabbit menisci	Meniscus regeneration	[149]
Decellularized adipose tissue	Human adipose-derived stem cells	-	Porcine adipose	Adipose tissue regeneration	[142]
Decellularized adipose tissue and alginate	Human subcutaneous preadipocyte cells	-	Human adipose	Engineering densely packed adipose tissue	[150]
Decellularized adipose tissue and plasma	Human dermal fibroblasts	-	Porcine adipose	Tissue substitutes with optimal microenvironment	[151]
Decellularized adipose tissue	Minced parathyroid glands	-	Porcine adipose	Maintaining biological functions of parathyroid glands	[152]
Decellularized skeletal muscle	Human skeletal muscle cells	-	Porcine tibialis anterior muscles and descending aortas	Volumetric muscle loss treatment	[153]
Methacrylate decellularized skeletal muscle	C2C12 cells	-	Lower limb muscle from adult Yorkshire porcine	Muscle regeneration	[154]
Decellularized liver tissue	HepG2 cells and human MSCs	-	Porcine liver tissue	Liver in vitro models for transplantation and drug screening	[155]
Decellularized liver tissue	Human iPSCs	-	Liver from three-month-old healthy Yorkshire pigs	Fabrication of patient-specific tissue substitutes	[144]
Decellularized liver tissue and collagen I	HepG2 cells	-	Porcine liver from Yorkshire pigs	Disease mechanism exploration and drug screening	[156]
Decellularized skin tissue	Human neonatal epidermal keratinocytes, human adipose-derived MSCs	-	Porcine skin tissue	Skin regeneration	[157]
Decellularized skin tissue	Human dermal fibroblasts	-	Porcine skin tissue	Dermal substitute	[158]
Decellularized skin tissue	Mouse fibroblasts	-	Native skin tissues from a Korean domestic pig	Establishing a 3D cell printing process	[159]
Decellularized cornea and collagen I	Human corneal keratocytes	-	Bovine eyeballs	Cornea substitutes	[160]
Decellularized cornea	-	-	Bovine eyeballs	Artificial corneas	[161]
Decellularized brain	Glioblastoma cells and endothelial cells	-	Cephalic parts of market pigs	In vitro disease model	[162]
Decellularized pancreatic tissues	Rat islets and endothelial cells	-	Porcine pancreatic tissue	Fabricating 3D pancreatic tissue constructs	[163]
Decellularized pancreatic tissues	-	-	Porcine pancreatic tissue	Pancreatic tissue substitutes	[164]
Decellularized tracheal mucosa	Endothelial cells and fibroblasts	-	Porcine tracheal mucosa	Functional airway-on-a-chip	[165]
Decellularized trachea	Human inferior turbinate MSCs	-	Porcine trachea	Tracheal reconstruction	[166]
Decellularized vascular tissue	Endothelial progenitor cells	-	Porcine descending aortas of pigs	Therapy for ischemic disease	[31]
Decellularized vascular tissue	Endothelial cells	-	Fresh porcine aortic tissue	In vitro vascular models	[167]

## Data Availability

Not applicable.

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
