# Peer review of "Recent Advances in Decellularized Extracellular Matrix-Based Bioinks for 3D Bioprinting in Tissue Engineering"

_materials, 2023, doi:10.3390/ma16083197_

Round 1

Reviewer 1 Report

This manuscript investigated decellularized extracellular matrix (dECM) bioinks for 3D bioprinting technology. Bioprinting technologies, bioactive molecules in the decellularized extracellular matrix, decellularization strategies, requirements, and recent applications utilizing dECM bioinks were described at a glance. However, there are some concerns to be published as listed below.

1. It would be better to describe leading-edge studies related to 3D bioprinting technology with dECM bioinks. The following articles will be helpful references. - Reference for section 6.6. Skin: https://doi.org/10.1016/j.biomaterials.2021.120776 - Reference for section 6.3. Adipose tissue: https://doi.org/10.1002/adfm.202200203

2. There are many typos and grammatical errors including, but not limited to, the title of section 6.11, 'Vascular.'

Reviewer 2 Report

Zhe et al. reviewed the recent advancement in decellularized extracellular matrix-bioinkh for 3D bioprinting in tissue engineering and biofabrication. This ms discuss the current status and various perspectives of decellularizied extracellular matrix bionics for bioprinting. Furthermore, different bioprinting modalities were discussed in this ms.
Overall, this is critical time for this review and it is well-written. After minor revision below, this ms can be accepted.

- 3D Bioprinting of dECM bioniks for drug testing and development should be summarized. (Peng, Weijie, et al. "3D bioprinting for drug discovery and development in pharmaceutics." Acta biomaterialia 57 (2017): 26-46.)

- Some of critical references are missing in this ms.
-Nam, Seung Yun, and Sang-Hyug Park. "ECM based bioink for tissue mimetic 3D bioprinting." Biomimetic Medical Materials: From Nanotechnology to 3d Bioprinting (2018): 335-353.
 Bioprinting ref
-Kang, Hyun-Wook, et al. "A 3D bioprinting system to produce human-scale tissue constructs with structural integrity." Nature biotechnology 34.3 (2016): 312-319.

Reviewer 3 Report

In the manuscript, "Recent advances in decellularized extracellular matrix-based bioinks for 3D bioprinting in tissue engineering ", the authors have discussed about dECM-based bioinks applied in 3D bioprinting. In addition, various bioprinting techniques and decellularization methods were also discussed in this study. The reported work has enough quality to publish in Materials after addressing the following comments.

Major comments

1.      The authors can make use of tables to represent the current in the field using the below titles;

i)                   Different forms of bioinks and bioink formulations

ii)                 Source of dECM

iii)               Application on what tissue etc  

2.      In section 6.3 the authors have discussed about only one single paper, they can add a few more and discuss it elaborately.

3.      Section 7 is not much convincing as too generalised  and can be improved to make it specific

4.      Authors should cite most recent articles wherever appropriate

Minor Comments

1.      In abstract the term tissue bioprinting” is quite unusual

2.      Type errors should be fixed for instance – line 247 whit  (with*) and line 586 In vivoImplanted (In vivo implanted*) - use of italics font and spacing.

3.      Several sentences throughout the paper are not in proper form and needs rephrasing such as line 404-405, 422-423

4.      In section 4 authors are describing various methods for obtaining dECM where they can even add recently introduced methods such as use of DME (liquefied dimethyl ether)

5.      In line 500 the authors have added “crosslinking of thiol-acrylate chemistry”, the need of adding chemistry is not necessary and if added it may be elaborated.

6.      Section 7 first paragraph does not have any connections with the heading of the section (Current challenges and further perspective)

7.      Repetition of words should be avoided and are found in many places throughout the article a few of them are as follows;

i)       use of although in line 685-686

ii)     use of on the other hand in line 712-714

Reviewer 4 Report

The reviewer received the manuscript "Recent advances in decellularized extracellular matrix-based 2 bioinks for 3D bioprinting in tissue engineering", prepared by a large team of authors.
The text of the manuscript is a good overview of the latest developments in the use of dECM for regenerative medicine, with an in-depth description of specific applications for the restoration of certain types of tissues using 3D printing.
I have a positive impression of the prepared review, I received new relevant information that is useful for my work from the point of view of a laser physicist, who also works in the field of laser technologies for regenerative medicine. I would recommend this review to my colleagues for reading.
However, I have a number of comments.
1) Do not consider me a bore, but I see a somewhat incorrect interpretation of the term "Laser-assisted bioprinting". Figure 1C shows a scheme for the formation of scaffold structures, which does not require the mandatory use of laser radiation. A large part of the bioprinting systems based on the photocuring of light-sensitive materials use LED light sources in the ultraviolet range. Therefore, in this case, I would apply Light-assisted bioprinting to this particular drawing.
2) In section 2.2. Laser-assisted bioprinting you provide information about a laser bioprinting method based on the principle of laser-assisted forward transfer, also referred to as LIFT (Laser Induced Forward Transfer). However, I don't see a picture of this method in figure 1 - I would suggest adding a diagram of it to this figure under item 1D. Also, I want to note that this method is primarily used as a method of spatial transfer of living cells, and it differs very significantly from the scaffolding method presented in Figure 1C. I would recommend that you add an additional paragraph in section 2 that describes the light-assisted bioprinting shown in Figure 1C.
3) Since I had experience with the supercritical CO2 technique for materials science problems, I think that you undeservedly told little about this technique. This is my subjective opinion, however, I recommend that you expand the description of this technique (I recommend https://doi.org/10.1016/j.msec.2017.02.002 ) by emphasizing its benefits. I would even recommend putting it in Figure 4.
4) I have some remarks about figure 5. A significant part of the readers of your manuscript will get acquainted with your work using a desktop computer. When studying this figure, the reader will have to tilt his head very much to both sides in order to read the side and bottom text. I would suggest flipping the text in the bottom half of the drawing. Please take care of your readers' necks.

In general, I want to repeat - I really like your manuscript, I find it extremely useful for myself and I am sure that it will also be useful to my colleagues.
I am sure that after a little correction your manuscript will be published.
